# Knot Formation on DNA Pushed Inside Chiral Nanochannels

**DOI:** 10.3390/polym15204185

**Published:** 2023-10-22

**Authors:** Renáta Rusková, Dušan Račko

**Affiliations:** Polymer Institute of the Slovak Academy of Sciences, Dúbravská cesta 9, 845 41 Bratislava, Slovakia

**Keywords:** polymer, DNA, molecular simulations, nanochannels, topology, knot, chirality, membranes

## Abstract

We performed coarse-grained molecular dynamics simulations of DNA polymers pushed inside infinite open chiral and achiral channels. We investigated the behavior of the polymer metrics in terms of span, monomer distributions and changes of topological state of the polymer in the channels. We also compared the regime of pushing a polymer inside the infinite channel to the case of polymer compression in finite channels of knot factories investigated in earlier works. We observed that the compression in the open channels affects the polymer metrics to different extents in chiral and achiral channels. We also observed that the chiral channels give rise to the formation of equichiral knots with the same handedness as the handedness of the chiral channels.

## 1. Introduction

Polymers are long molecules consisting of many building units called monomers [1]. The units can be chemically identical, and still it is possible to think of an infinite number of polymers that could be constructed and yet differ by only how the monomers are connected. As such, the polymers represent a combinatorial problem that is suitable for studying by computer simulations.

The way the monomers are connected defines the polymer’s topology. A relatively new topology is represented by polymer knots [2]. The knots are formed by winding a long polymer chain around itself. The polymer knots occur naturally both in synthetic and biological polymers, such as DNA and proteins. The first synthetic knots with controlled topology were made possible by the end of the 1980s [3]. Knotted polymers have been drawing increasing scientific attention given the progress in macromolecular synthesis, biology, mathematics, and molecular simulations.

It is difficult to pinpoint when and how this scientific interest in knotted molecules began, whether it was sparked by imagination or, as is often the case, by observing nature. But now, it is clear that the topological state of molecules has strong biological and technological effects and implications. While in biology, the knots can be very harmful on a genome [4,5,6,7,8] but very important on proteins [9,10,11], whose biological function is yet to be fully revealed, the distinct effects of polymer knotted topology currently pose problems that need to be engineered due to a current lack of polymers with a well-defined knotted topology and methods to synthesize them in sufficient amounts.

As pointed out above, polymers, especially with regard to their topology, represent a problem suitable to be studied using computers. While the progress in polymer synthesis enabled controlled preparations of knotted molecules of up to eight crossings [12], in computer simulations, knotted polymers with well-defined topology are easily prepared. Thus, computer simulations are currently a useful and indispensable approach in investigating knotted polymers.

As we also mentioned, the knots are formed naturally by biological processes [13,14], but are also inevitable in physical processes, when the probability of a polymer being knotted increases with its length [15]. The simplest and most practical way to produce knots would be using a diluted solution of long polymers and ligating the ends of the polymer. This would, however, lead to the production of a wide spectrum of knots. Computer simulations are useful to provide a prediction of how the length of polymers, such as DNA, and ionic environments would be used to control the topology of the polymer knots [16,17,18].

The next layer of complexity is added by investigating polymer knotting in confined spaces. The confinement state is the most typical state where polymers occur in nature [19,20]. Confinement is a state where the natural dimension of the polymer is larger than the size of the confinement geometry. Biopolymers, such as DNA, are naturally confined in such small spaces as viral capsids or cellular nuclei. The state of polymer confinement is also essential for nanotechnology, including nanocomposite or nanofluidic experiments involved in genomic studies. The confinement of polymers, yet while a very complex state, being a subject of experimental, theoretical, and simulations studies, is known to enhance and stabilize the knottedness of polymers [17,21,22]. Computer simulations provide insights beyond the possibilities of experimental imaging.

Consequently, the level of the complexity of the problem can be extended by adding external forces into consideration that induce compression of the confined polymer. The compression of polymer chains under confinement was studied by means of Monte Carlo (MC) [23,24,25] and molecular dynamics (MD) simulations [26,27,28,29,30,31,32,33,34,35] in channels with square [23,31,32,33,34], cylindrical [24,25,27,28,29,30,35], or helical [36] sections or in structured channels [26,37,38], where the compressive force was applied by pulling on distant ends of the confined polymer in a direction against each other [23,24,27], by using a piston compression similar to the gaskets used in the experiments described in [24,25,28,29,31,32,33,34,35,36] or by flow of media [26,37,38], while compression in a spherical confinement was also investigated [39,40]. While previous research has generally highlighted the enhanced effects of compression in generating entanglements within polymers, only a few studies have investigated the topology of compressed polymers [28,36,40]. These works quantified knot complexity and the populations of knots as a function of compressive force.

It has been demonstrated that the theoretical and computational insights into polymer compression within nanochannels can be experimentally validated by confining DNA within nanofluidic channels and inducing compression through specific experimental setups [41,42]. More recently, the earlier computational findings regarding knot formation in confined spaces, along with developed nanofluidic experiments, have led to the creation of the ‘knot factory’ nanofluidic device. This innovative device produces knots when polymer chains are compressed inside nanochannels [43]. Furthermore, computer simulations have been instrumental in exploring various aspects of this device, including the compression duration and the compressive force’s impact on knot topology [28]. Our own computational work has also contributed by investigating the effects of nanofluidic channel sizes and the strength of confinement [36]. The computer simulations carried out by us [36] extrapolated the model [28] used previously to model the experimental setup in [43], and also devised an in silico experiment to test whether the chiral geometry of the channels can induce the handedness and influence the chirality of knotted structures. 

Chirality is a prominent property of knots. As mentioned above, knots are formed by a substantially long polymer chain winding around itself. One of the parameters to characterize knots is the crossing number that quantifies how many times the polymer winds around itself. The direction of the polymer winding around itself defines the chirality of the knot. The crossing number is a combinatorial property, and the number of possible knot types that can be constructed with a given crossing number increases substantially. This is why it is feasible to create as many as 1.7 million prime knots with up to 16 crossings, out of which fewer than 2000 knots are achiral [44]. The chirality of polymer knots exhibits unique physical properties. However, due to the limited availability of polymers with well-defined knotted topologies, practical applications are currently largely theoretical. Nevertheless, experimental evidence suggests that chiral knots play a role in biology [45], can be employed to control optical properties [46], offer potential for new energy-harvesting sources at the nanoscale [47], and they may find applications in stereoselective chemosensing [48] and also in the progress in organized entanglements in chemistry [49]. 

In the context of knot formation, a recent computer simulation was devised to explore an intriguing scenario of whether and how knotting could be induced by simply pushing DNA inside nanochannels, without the need for the more complex lab-on-chip nanofluidic experiments [34]. This scenario holds practical significance because it offers a relatively straightforward way to induce knot formation in polymers, which has relevance in various applications where polymers are pushed through narrow channels, such as in chromatographic resins or membranes. Additionally, experimental efforts are underway to develop chiral membranes capable of separating chiral enantiomers based on their geometric properties, which includes the use of structures like helical nanochannels [50,51]. The emerging applications of chiral knots mentioned above together with chiral separation devices are part of the emerging field of chiral nanotechnology [52], where our computer simulations contribute by proposing methods for producing knotted structures while controlling chirality through physical means.

In our current study, we are exploring the formation of knots in polymers as they are pushed into open, infinitely long nanochannels with varying sizes and geometries. To simulate chiral environments, we designed these channels with a helical geometry and induced different chirality by altering the winding direction of the helical loops within the nanochannels. We developed a novel computational approach to identify the chiral properties of the knots that form in the DNA strands as they are pushed inside these channels. Our method utilizes Knoto-ID [53], a topological software, to determine the chirality of the knots in reference to the Rolfsen knot table [54]. Furthermore, it identifies handedness based on a new, biologically motivated knot table [55]. We compare and discuss the simulations also with respect to the compression in finite channels by pushing against an impenetrable wall developed and modeled in Ref. [36]. The structure of this manuscript is as follows: in Section 3.1, we present the results concerning polymer metrics; in Section 3.2, we discuss various aspects of monomer distribution; Section 3.3 explores knotting probabilities; Section 3.4 introduces the computational routine used to analyze knot handedness, and, finally, in Section 3.5, we discuss the possible mechanism through which the chiral environment impacts the handedness of knots. We believe that our findings significantly contribute to our current understanding of polymers in confined spaces, especially under compressive forces. These insights can have implications in the fields of DNA biology, polymer physics, and the emerging field of chiral nanotechnology.

## 2. Materials and Methods

### 2.1. Model of DNA 

The dsDNA is modeled as a discretized beaded chain consisting of N = 300 beads representing DNA portions with a width of 1 *σ* corresponding to 2.5 nm [56]. The beads interact via bonded and nonbonded interactions. The bonded interactions are represented by covalent bonds modeled by bond-stretching and angle-bending potentials. The bond-stretching is modeled by a harmonic potential in the form *U*_S_(r) = *K*_s_(r−r_0_)^2^, where r is the position vector of the bead, r_0_ is the equilibrium value set to r_0_ ≡ *ℓ* = 1 *σ*, and *K*_S_ is the penalty against bond stretching. In order to prevent artificial strand passages under compressive forces of confinement and piston compression, the force constant *K*_S_ was set to 80 *ε*_0_, where *ε*_0_ = k_B_*T* represents energy of thermal fluctuations. The angle-bending interaction is modeled by a harmonic potential in the form *U*_b_(*θ*) = *K*_b_(*θ* − *θ*_0_)^2^, where *θ* is the angle between vectors of two consecutive monomers in the chain, *θ*_0_ is the equilibrium angle set to *θ*_0_ = π, and *K*_b_ is the force constant of the interaction representing the energy penalty against bending of the angle. The force constant *K*_b_ relates to the persistence length of the DNA molecule and we set the value *K*_b_ = 20 *σ*/*ε*_0_, thus imposing on the beaded chain a common value for the persistence length of the DNA molecule [57]. The nonbonded terms of the potential involve excluded volume interaction. The excluded volume interaction describes the volume occupied by monomers of the chain and it was modeled by fully repulsive cut and shifted Lennard-Jones potential *U*_ex_(r_ij_) = 4*ε*_0_[(*σ*/|r_ij_|)^12^ − (*σ*/|r_ij_|)^6^] + 1/4 if |r_ij_| < 2^1/6^*σ* and *U*_ex_(r_ij_) = 0 otherwise, where |r_ij_| is a distance between a pair of beads position vectors r_i_ and r_j_, where *i* ≠ *j*.

### 2.2. Model of Nano-Channels

The channels were modeled by using an implicit helical confinement with a helical geometry developed in the previous work [36]. The channel is modeled as a tube with radius *R*_ch_ and central axis described by the equation of a helix given as $r_{0}\left(t\right)=kt\hat{i}+R_{H}\cos\left(wt\right)\hat{j}+R_{H}sin(wt)\hat{k}$, where *t* is a periodic parameter in radial space and ω gives a subtended angle as *t* increases [58]. The parameter *ω* also carries information on the handedness of the channels, where *ω*/|*ω*|<0 corresponds to left-handed channels with negative winding and *ω*/|*ω*|>0 is used to model right-handed channels with positive winding. The channels are defined by four parameters: sign(*ω*), radius of the channel *R*_ch_, radius of the helix, *R*_H_, and the pitch, *k*. The pitch of the channel determines the distance between the helical loops of the channel, *d*_H_ = 2π*kσ*. If the radius of helix is set to *R*_H_ = 0, the geometry of the channel corresponds to a simple cylindrical channel regardless of the settings of *k* and *ω*. The model for implicit helical confinement was implemented into the Extensible Simulation Package for Research on Soft Matter Systems (ESPResSo v. 4.2) software that was used in the simulations [59,60]. The model for the implicit helical confinement implemented an algorithm for calculation of the distance of a point from the helix [61]. Despite the algorithm involving an iterative step, we experienced that the simulations are very fast and stable for a wide range of settings tested in the current and the previous work [36]. Within the simulations, we simulated DNA chains in the channels with the radii of the channel corresponding to three confinement strengths given in terms of the ratio of the channel diameter to the polymer’s persistence, *D*/*P* = 2*^i^*, where *i* = −1, 0, and 1. The corresponding diameters of the channels in physical units correspond to 10, 20, and 40 nm. These sizes of nanochannels are relevant to genomic experiments [62] and are also achievable in the preparation of chiral membranes [50]. As for the particular parameter settings, the radius of the helix was set to *R*_H_ = 1/3 *R*_ch_, based on our previous work, where the effects of the chiral confinement were the strongest in the range of *R*_H_ = 1/3 − ½ *R*_ch_ [63]. In the case of the simulations of cylindrical channels, the setting of *R*_H_ was 0. The pitch was set to *k* = *D*/2π to allow for comparisons and discussion of differences between simulations of chains pushed into open infinite channels and those compressed in the blinded nanochannels of knot factories investigated recently [36]. Both of the possible chiral scenarios were simulated with the handedness of the channels being *ω* = −1 and 1. The polymer was inserted into the channel as a stretched chain with beads placed along the major axis of symmetry of the channel corresponding to the *x*-axis, with initial coordinates r_i,x_ = *I*, where *i* = 1…*N*, and r_i,y_ = r_i,z_ = 0. 

### 2.3. Push by External Force and MD Simulation

The push of the DNA chains was intermediated by a piston modeled by an additional bead with a very large radius, i.e., a radius of the piston *R*_P_ >> *R*_ch_. The “piston bead” moved along the main axis of symmetry of the channel corresponding to the *x*-axis, while the movement in the *y* and *z* directions was constrained. The piston bead also interacted with the chain only by the excluded volume interaction in the form, as provided above, and with the setting of *σ*_P_ = 100 *σ*. We applied external force on the piston bead with the range of values *Fσ*/ε_0_ = 0.1, 0.5, 1, 2, and 5. The region of interest for these forces was chosen based on previous studies [24,25,36] while omitting the special regime of very small forces in narrow channels. The pushing forces were transformed to velocities of pushing after performing the simulations and computing the piston velocity as the overall distance traveled by the piston over the period of simulation time, *v* = *d*_T_/*τ*_sim_. The pushing of the DNA inside the channel was carried out as in the recent molecular simulation work by using Langevin dynamics [34]. We carried out Langevin molecular dynamics simulations by solving Langevin equations of motion for each bead $m\ddot{r}=-\nabla U\left(r\right)-\gamma m\dot{r}+R\left(t\right)\sqrt{2\varepsilon_{0}m\gamma }+F_{ext}$, where each term in the equation represents the force acting on the bead, ∇U(r) is given by the molecular potential, $-\gamma m\dot{r}$ represents the friction, and $R\left(t\right)\sqrt{2\varepsilon_{0}m\gamma }$ is the random kicking force, where *R*(*t*) is a delta-correlated stationary Gaussian process. The last term in the equation is the added external force and applies only to the piston bead with *F*_ext_*σ*/ε_0_ = *Fσ*/ε_0_ = 0.1, 0.5, 1, 2, and 5. The Langevin equation does not consider hydrodynamic interactions between monomers. For simulations involving out-of-equilibrium systems, especially when intense interpenetrations and interactions among polymer segments are present, a more suitable thermostat, such as dissipative particle dynamics (DPD), is required. This is notably encountered in bulk polymer brushes [64]. However, results from the Langevin dynamics and DPD converge when simulating systems with a low extent of monomer interactions, typically at concentrations below the collective regime [65]. For instance, recent work employed Langevin dynamics to simulate the pushing of DNA inside nanochannels [34]. This choice was justified by the fact that the chain conformations reside in the transition region between the Odijk and deGennes regimes where the hydrodynamic interactions are effectively screened and the collective regime does not apply [66]. 

The equations of motion were integrated with the step size Δ*τ* = 0.010. The optimal size of the integration step Δ*τ* was determined through previous simulations [36,63]. It is small enough to maintain the stability of the equations of motion during integration, while also maximizing the performance of computer simulations and the usage of computer time. The coarse-grained time is transformed to *τ* = 6*πησ*^3^/*ε* physical units [56], where *η* is the viscosity of media. After the chains were inserted into the channel with the initial coordinates r_i,x_ = *i*, where *i* = 1…*N*, and r_i,y_ = r_i,z_ = 0, we performed an initial pre-equilibration run with 10^7^ iterative steps. Afterwards, the main simulation started with 10 repeated runs for each parameter setting performing 10^9^ integrations, while we were also collecting data for analyses of polymer metrics, monomer distributions in the channels, and topological analyses of polymer knottedness. 

## 3. Results and Discussion

### 3.1. General Polymer Metrics

First of all, we characterized the simulated systems by evaluating basic polymer metrics. Polymer metrics provide first and important information about polymer conformation, its size, and polymer behavior in the presence of confinement and compressive forces, realized here by a pushing force mediated by a piston or gasket. Polymer metrics also allow the bridging of gaps in standing theoretical understanding and gaining of insights into how the confinement and presence of external forces alter the behavior of the polymer. 

Based on our previous experience from our previous study [36] and also in context of existing works relevant to studying polymers under compression in nanochannels [24,25,30], we chose to evaluate the polymer metrics in terms of polymer chain span, although the polymer metrics in terms of end-to-end distance and gyration radius are provided in Appendix A. Given a certain configuration, the span is calculated as the distance separating the two farthest beads of the chain. 

The span is calculated as the maximum distance between two monomers, represented by coarse-grained beads, that can be found on the chain. The distances are calculated by using position vectors of the beads. When computing the span, one can use all Cartesian coordinates or compute the span only using the coordinate along the major axis of inertia of the channel. We decided on the latter case, since it diminishes chain size effects. The span is defined as *S*(*x*) = max|**r**_i,x_, **r**_j,x_|, where *i* ≠ *j* and *i* ϵ 1…*N*. The computed span is shown in Figure 1. In Figure 1, we not only show and compare span as obtained on polymerspushed inside the open infinite channels with helical and cylindrical geometry, but we took also advantage of having studied the case of polymers compressed by a gasket in blinded channels with an impenetrable wall at their bottom, corresponding to the experimental setting of knot factories [36]. 

Figure 1a shows the dependence of the polymer’s span, as obtained in channels in helical and cylindrical geometries, for three confinement strengths expressed in terms of the ratio of the diameter of the channel to polymer persistence length, *D*/*P* = 0.5, 1, and 2. The values of the span are shown as black lines for helical channels and red lines for cylindrical channels, as also illustrated by inset snapshots from the simulations. The filled area indicates differences of span obtained for a given setting confinement strength *D*/*P* and between the two investigated geometries of the channels. 

The span is shown as a function of velocity of the piston pushing the polymer along the infinite channels. The push by the piston is realized by applying an external force to the piston. Hence, the velocity of the piston was obtained from the simulated trajectories as a total distance traveled by the center of the piston from the beginning of the simulation to its final position at the end of the simulation over the simulation time *τ*, *v* = |**r**_P_(0),**r**_P_(*τ*)|/*τ*. The values of the velocities, the polymer spans, and other data evaluated later in the work were averaged over ten simulated trajectories. The obtained velocities are shown in the inset graph as a function of applied external force. The velocities *v* range from 1.2 × 10^−4^–6.6 × 10^−3^ *σ*/*τ*, while the physical dimensions of the units are [*σ*] = 2.5 nm and [*τ*] = 74 ns × (*η*/*η*_0_) ns [56], where *η*_0_ is the viscosity of pure water, *η*_0_ = 1 cP, and *η* is the viscosity of the actual buffer used in the nanofluidic experiment. The buffers in nanofluidic applications often consist of a solution containing polymers, saccharose, agarose, etc., to increase the hydrodynamic drag of the media on the molecule [41,42], while viscosity can be increased to 10–80 cP [67]. The correction for the viscosity gives some space for variation to transformed value of the physical time units; nonetheless, the values of experimental velocities of pushing, in the order of μm/s, are accessible. 

We would like to note that we opted for realizing the push intermediated by an applied external force instead of directly moving the piston by a constant distance at a time, as simulated in some existing works [34]. This allowed us to directly compare the metrics of the polymer pushed in open infinite channels to the previously obtained results on polymers compressed by the external forces inside channels that were blinded by an impenetrable wall [36]. 

The comparison of polymer metrics pushed in helical and cylindrical geometries on Figure 1a shows that there is a significant difference between the chain span observed in very narrow channels, with *D*/*P* = 0.5. The difference seems to be higher than in the compression in channels with an impenetrable wall. The difference in polymer span also seems to disappear at weak confinement strengths in terms of *D*/*P*, especially if strong compressive forces are applied.

In Figure 1b,c, we show the computed polymer spans as obtained in channels with helical and cylindrical geometry. Figure 1b compares the case of polymers pushed inside infinite open channels to the case of polymers compressed in channels with an impenetrable wall and helical geometry. Figure 1c, on the other hand, compares the pushing versus compression in channels with cylindrical geometry. The data for the current simulations are shown as black lines, and they are compared to the simulations in blinded channels shown as red lines. The insets show the simulation settings by snapshots taken from the simulations augmented by hatching schematics. 

The comparison of the computed span shows, in general, smaller compaction of the polymer when compressed by pushing inside the open channels than in the case when the polymer is compressed against the impenetrable wall in the nanochannels. We understand the obtained results as follows. Given the form of the equations of motions in the Langevin dynamics, provided in the Methodology section (Section 2.3), the hydrodynamic drag force, represented by $-\gamma m\dot{r}$ in the Langevin equation, opposes the motion of the particle, leading to a damping effect. The strength of this damping is determined by the value of *γ*. As the external force acting on the particle increases, the hydrodynamic drag force remains proportional to the velocity of the particle ($-\gamma m\dot{r}$). The linear relationship between hydrodynamic drag and external force is confirmed by the computed velocities as a function of external forces, shown as inset in Figure 1a. When applying compressive force to a polymer confined in an open channel, some of the compressive energy is dissipated by the movement of the chain through the media. As the force-to-displacement ratio still follows the established relationship—*F*·*D* ∝ *S*^−9/4^ (with *F* corresponding to force, *D* is the diameter of the channels, and *S* is the span) [30], this means that opening the channels acts like compressing the polymer with smaller force. 

We also showed in the previous work that the helical confinement in narrow channels acted to a certain extent like cylindrical channels with a smaller diameter, i.e., channels with higher confinement strength. The conformation of the DNA molecule is determined by a balance of several ongoing forces: the confinement force, elastic force, and hydrodynamic force intermediated by the pushing force [38]. This is a complex relation, where we cannot directly compare the obtained data to a predictive model, but we may bridge the theoretical understanding with our computer experiments. In Figure 1b,c, the computed polymer metrics in terms of polymer span show not only that there are differences between compression in open infinite channels and compression in the nanochannels against an impenetrable wall, but the extent of the differences in DNA compaction is significantly influenced by the geometry of the channels. 

As investigated in the existing studies, the compaction of the polymer under external forces is relevant and related to conformational changes [24,25,36], and it is important for devising a control mechanism for the topological state of the polymer for nanotechnological applications [28,34,36,43]. The data also indicate that compression by pushing the polymer into open channels, as compared to the case of compressing the polymer against an impenetrable wall inside finite nanochannels, is much less effective, especially in the case of the helical nanochannels, which will lead to lower degree of knotting, at least at the current setting of the helical geometry in terms of the pitch of the channels. In addition to the data in Figure 1b,c, it is noteworthy that at strong confinement forces in terms of *D*/*P* and small compressive forces, the existence of a special regime was discovered, forming a shoulder on the dependencies of the span versus compressive forces [24,25], that was not properly captured, as we did not probe the region in detail by applying a range of sufficiently small compressive forces. The reason is that there are only minor topological differences throughout this region, with the polymer being mostly unknotted; hence, the region is not in the focus of this work.

### 3.2. The Monomer Distributions upon Pushing

The polymer metrics provide one-dimensional information on the polymer behavior and the effects of confinement and compressive force. Another convenient property that is directly accessible from computer molecular simulations to represent the situation of a dynamically moving polymer molecule is distributions of monomers. Here, we analyze the radial and axial distributions of monomers across the major axes of inertia of the confining channels. 

Figure 2 is a composite figure that shows the situation of the polymer’s monomers within the channel, providing complementary information to the polymer metrics evaluated in the previous Section 3.1. We believe the figure, as presented, provides advantageous insight for readers when convening to a concise explanation of the figure. Figure 2 is divided into six panels. Each panel shows the heatmaps of the monomer distributions inside the nanochannel at the very left side of the panel. The heatmaps present valuable pictures of the overall distribution of monomers. Since the monomers travel through the channel, sometimes to very long distances, during pushing, we modified the method of calculating the heatmaps employed in earlier works [36]. The heatmaps show concentrations of monomers through the periodized distance of one helical turn, *d*_H_ = 2π*kσ*, where *k* = *D*/2π in helical channels, and *d*_H_ simply equals *D*. There are five heatmaps corresponding to five settings of the external force, from the bottom up following the increasing velocity of pushing, as shown in the inset of Figure 1a. 

The heatmaps indicate expulsion of monomers into the lateral sides of the channel with an increasing velocity of pushing. This effect is numerically captured in the graphs showing radial distribution functions that are displayed adjacent to the heatmaps in each of the panels. The radial distributions show a similar shape to those already obtained for cylindrical [36,68] and square channels [69], with a maximum of the number density of monomers in the middle of the channels. As the velocity of pushing increases, the radial distribution functions flatten, and the distributions show a drop in the monomer concentration in the center of the channel. The direction of the concentration drop with increasing velocity of pushing is indicated by a downwards arrow in the part of the distribution corresponding to the center of the channel. At the same time, the monomers are pushed and redistributed towards the lateral sides with increasing velocity of pushing. The increasing number concentration, expressed as a normalized frequency of occurrence, at the lateral sides of the channel in the direction of increasing velocity of pushing is indicated by an upwards arrow.

To the right of the radial distributions on each panel, we also show the axial distribution of monomers along the main axis of inertia of the channels. These distributions show the number density or concentration of monomers from the position of the piston. In the axial distributions, the position of the piston is always at the origin. The values on the *x*-axis also reflect the direction of push in the simulations, which went from right to left in Cartesian coordinates. The shape of the concentration profiles is very similar to what is observed experimentally in the dynamic nonequilibrium segmental concentration profile of a single nanochannel-confined DNA molecule [42,43]. In our case, the profiles correspond to equilibrium profiles obtained for different velocities of pushing. The axial distribution shows an evolution with an emerging peak in the number density of monomers near the surface of the piston. The maximum of the distribution increases with increasing velocity of pushing. At the same time, the distribution or occupancy of the channel along the major axis of inertia in the direction of pushing becomes narrower. This narrow region can be associated with a conformational transition with increased spooling [34,35,70], identified earlier to occur under large compressive forces. The insets in the graphs with axial distribution of monomers also show positioning of monomers along the channel in terms of their bead index as a function of their coordinates along the main axis of inertia (*x*-coordinate). This kind of projection was used in some existing works studying polymers in nanochannels by other authors [34,35]. The insets show the positionings for two limiting cases of the velocity settings used in the current simulations, i.e., *Fσ*/*ε*_0_ = 0.1 and 5. Each panel also shows a representative snapshot from the simulation showing polymer in the channel with a particular geometry obtained at the end of the simulation for the setting of *Fσ*/ε_0_ = 1.

Figure 2a–c show the distribution of monomers for cylindrical channels, and Figure 2d–f show the information as obtained in the channels with helical geometry. The rows show computed distributions for different settings of confinement strength, indicated on the graphs as *D*/*P* = 0.5, 1, and 2. We can see that in the absolute numbers, a larger maximum on the axial distribution near the position of the piston surface is achieved in the case of cylindrical channels. Also, the observed flattening of the radial distributions is more extensive in the case of cylindrical channels. This indicates that when pushing the polymers inside open channels, the force is less effective in compressing the polymers, and the monomers do not fully explore the helical grooves of the helical channels. 

Consistently with the polymer metrics shown in Figure 1, the data indicate that the extent of compaction is lower than that previously observed for finite channels with an impenetrable wall [36]. The previous studies showed that the level of compaction with higher compressive forces applied is directly related to the extent of knotting [28,34,43]. In our previous simulations, we also saw that helical geometry by itself enhances knotting as compared to simple cylindrical geometry. In the current simulations, the lower level of compaction and lower effectiveness of compression with given forces in open channels suggest that smaller levels of knotting will be expected, especially for very narrow channels and strong confinement. 

### 3.3. Knotting Probabilities and Topology

In this section, we evaluate the topological state of DNA polymers under compression induced by pushing through open infinite channels. The knotting probability is evaluated as the frequency of finding knots in ten runs along the trajectories that contain 5000 structures each for topological analyses. The occurrence of knots was evaluated by Knoto-ID software [53] (v1.3.0), which uses Jones’s polynomial, allowing us to obtain information on the handedness of the knots. The knotting probability analyses focused on obtaining information on the complexity of knots in terms of the crossing numbers, knot groups evaluating presence of amphichiral knots, twist knots, torus knots, unknots, and unidentified knots with a crossing number larger than 11. Finally, the analyses also investigated the chirality of knots in terms of the occurrence of right-handed and left-handed knots.

Figure 3 presents and summarizes the findings about the topology of the polymers. The first two columns compare the knotting probability as a function of the velocity of pushing or compressive force. The arrows above the columns indicate the direction of the increasing velocity of pushing. The frequencies of occurrence of knots with a given complexity characterized by their crossing numbers are shown as stacked areas distinguished by colors in a thermometer scale. The crossing numbers corresponding to particular colors at the employed scale are indicated at the bottom. The thermometer scale goes through a spectrum of colors from plain blue to plain red, where the plain blue corresponds to unknots and the plain red areas show presence of very complex knots with a crossing number larger than 11. The rows correspond to the particular setting of confinement strength, indicated in terms of the *D*/*P* ratio. 

The analysis shows that, in general, the probability of knotting increases with the velocity of pushing. Also, it is shifted towards the occurrence of more complex knots with increasing velocity of pushing. This observation is consistent with previous investigations of knotting in DNA pushed through square channels [34] and also in studies of polymers under compressive force in cylindrical [28,36] and helical channels [36]. The knotting probability also depends on the diameter of the nanochannels. Here, the dependence is similar to the situation of knots compressed in finite channels, where the knotting probability is the largest in the channels with *D*/*P* = 1. 

On the other hand, the distinctive feature of the pushing inside the infinite channels seems to be apparent lower knotting probability in nanochannels with helical geometry as compared to the cylindrical channels. As discussed in Section 3.1, when pushing inside the infinite channels, the resulting compaction is determined by establishing a balance between several forces, i.e., the confinement force, elastic force, and hydrodynamic force intermediated by the pushing force. The resulting difference in compaction observed by means of molecular simulations between finite and infinite channels indicates that opening the channels affects the compaction in channels with helical channels more than in the case of cylindrical channels. For this reason, unlike the previous simulations in finite channels, the helical channels do not enhance the knotting probability. 

The decreased knotting probability in helical channels can, however, be due to the specific geometric parameters of the helical channels determined by the pitch *k* = *D*/2π, which determines the distance between helical loops or size of the helical turns, *d*_H_ = 2π*kσ*. It is important to note that the current setting of the pitch was chosen based on our previous work where we investigated chiral effects in terms of mobility of localized knots with a given chirality. It is probable that for the current experimental setting of the polymers pushed inside the open channels, the pitch has to be fine-tuned, perhaps towards larger values above the deflection length *λ* = (*D*^2^/*P*)^1/3^ [71], so that *d*_H_ > 2π*kσ* for *k* fixed to *k* = *D*/2π. 

In order to gain insight into this behavior, we simulated DNA polymer pushed into a helical channel with the size of helical loops well above persistence length, set to *d*_H_ = 2 *P*, and analyzed the knotting probability, as shown in Appendix A. Appendix A surprisingly indicates significant compaction associated with high levels of knotting even at strong confinement forces (*D*/*P* = 0.5) and very small compressive forces (*Fσ*/*ε*_0_ = 0.1). Since detailed refinement and thorough exploration of the parameter settings of the channels’ geometry are clearly beyond the scope and extent of a single work, we will readdress it in future works, providing more information on polymer behavior for various parameter settings of the helical channels, polymer chain length, and persistence lengths.

The midsection of Figure 3a for helical channels also shows knotting probabilities for different knot groups, represented by amphichiral, torus, twist knots, unknots, and complex knots above 11 crossings for which the notation is not included in the topological software. It has to be noted that the populations are normalized again, but some of the groups overlap, such as the trefoil knot, which is both the twist and torus knot, and the 4_1 knot, which is both the twist knot and the amphichiral knot. The population of the amphichiral knots increases with the pushing velocity and the level of compaction under the compressive force of pushing, which increases the complexity of entanglements. 

Note that there are only 20 amphichiral knots out of 801 knots that can be constructed from knotted lines up to 11 crossings [72]. The populations of twist knots seem to be increasing in the case of strong and weak confinement forces, *D*/*P* = 0.5 and *D*/*P* = 2. This might be related to the fact that for both of these settings, the knottedness is lower than in the case of *D*/*P* = 1, and the amounts of existing twist and torus knot types as a function of crossing number do not evolve equally; in other words, with increasing crossing number, there are more twist knots than torus knots. For the cases outside intermediate confinement, *D*/*P* = 0.5 and 2, we see an abundance of unknots. In the case of the strong confinement, *D*/*P* = 0.5, the unknots can be related to lower degrees of compaction and prevailing effects of confinement keeping the chain extended. In the case of weak confinement, *D*/*P* = 2, chain length effects or timescale and velocity effects might be taking place. The polymer at its given length is much more diluted; hence, on one hand, it leads to much more spooling, but also it might have not enough time to explore the geometrical spaces of the larger channels at the fixed rate of pushing. This may lead to the higher extent of writhing indicated in Figure 3b, and perhaps much smaller forces/velocities of pushing should be investigated in the case of weak confinement in the future to obtain a general picture on polymer behavior in channels with geometric modulation. Such regimes of very slow pushing in wide channel will require separate computer experiments, given the heavy computational expenses inevitable for such a computer experiment.

### 3.4. Handedness of the Channels and Knots

We further investigate whether the effect of the handedness of the helical channels on the knot chirality is preserved to some extent, and if channels with helical geometry and given handedness can be used to control handedness of the knots that are created during the pushing of the DNA through the channels. In the previous study [36], we probed the chirality of the knots by computing writhe of the knotted part additionally to the topological analyses by KymoKnot software (http://kymoknot.sissa.it/) [73]. 

In the current work, we directly use the information on the chirality of the knots provided by Knoto-ID software [53]. However, the information had to be pretreated before the data could be used to compute statistics of left-handed and right-handed knots. Knoto-ID identifies the knot types by evaluating Jones’s polynomial and compares the identified knots with the notation in Rolfsen’s table of knots [54]. If the knot has the opposite chiral projection than shown in Rolfsen’s table, Knoto-ID identifies such knots as “mirror images” and indicates the chirality by the letter “m” in the name of the knot type in the result. Originally, Rolfsen’s table does not distinguish the knots based on their chirality, so knots with positive and negative writhes are mixed. This situation motivated Vázquez et al. to suggest the creation of a new biologically motivated knot table, where the aspect of the knot’s handedness in terms of the prevailing knot’s writhe would be considered and reflected [55]. In their work, the authors identified the knots with opposite chirality than indicated in the Rolfsen’s table. We used this sorted information together with the knot types detected by Knoto-ID software as Rolfsen’s analogues or their mirror images in order to identify the left-handed and right-handed knots. 

Moreover, mathematically, knots occur only on closed curves, and the algorithms for finding knots often involve some kind of closure method that constructs a connection between the free ends of the linear polymer chain. In order to eliminate the possible bias coming from the closure method, we evaluated only the knots found in conformations with an arbitrarily chosen very short end-to-end distance. It is noteworthy that although we do not know how the end-to-end distance and the bias from the closure method are related quantitatively, one intuitively expects that the number of entanglements introduced by closing the arc grows with the distance spanned by the added closing segments [74]. If we consider, for simplicity, a case where the end-to-end distance is equal to the size of the bond length *ℓ* = 1 *σ*, there would be no need to construct a closure. 

Furthermore, evaluating knots at short end-to-end distances can be of practical relevance, as the knots could be chemically embedded in the polymer by closing the polymer ring chemically. The distance was set to 10 *σ* based on the average variation of the end-to-end distance found in consecutive frames in simulated trajectories, and we consider it a ligation distance. After computing numbers of right-handed and left-handed knots, we evaluated their statistics, which are summarized in the last column of Figure 3. Here, the knots with the same handedness as the handedness of the helical channel are summed up as the number of equichiral knots. In the opposite case, the knots with the opposite handedness to that of the channels are summed up as antichiral ones. The graphs in the last column of Figure 3 show the ratio of the populations of the equichiral and antichiral knots. The graphs indicate that even in the regime of DNA compression by pushing inside the nanochannels, the ability of helical channels to give rise to equichiral knots is preserved, as we observed in the case of compression knot factories with finite nanochannels. It should be noted that in the new rigorous table by Vázquez et al. [55], chirality of knots was determined only up to nine crossings, which we used in our analyses and the plots in Figure 3 (others were not included); hence, the resolution of the chiral channels can be even higher but it is beyond current knowledge and is a subject for future investigation.

### 3.5. On the Mechanism of How Geometry of the Channels Induces Handedness of Knots

In addition to the information on knotting statistics, computer simulations can also help understand the mechanisms by which knotting occurs [28,75]. In the following, we propose a mechanism for how the chirality of the channels may control the chirality of knots and entanglements created on a DNA chain exposed to compression by being pushed inside the nanochannels. 

An earlier work investigated this by inspecting positions of emerging knots along the polymer chains [28], and found that the knots were created mainly by backfolding and threading of polymer ends through the loops. Another work also suggested that the knotting mechanism may involve maintaining contacts of the DNA polymer at specific sites, as found on ribosomal surfaces [75,76]. We believe that the process of knotting in chiral channels involves both compression-induced backfolding and threading, as well as the direction of writhing induced in polymer chains that are in contact with the walls of the chiral channels, aligning the polymer with the winding direction of the helical channels. 

The occurrence of backfolding is evident from the evolution of the polymer metrics in terms of the chain span, investigated in Section 3.1 and shown in Figure 1. It is also evident in the index-position projections of the monomer distributions, which we investigated in Section 3.2, and that show multiply folded conformations in Figure 2. A multiply folded chain creates loops that preferably turn and twist in the direction controlled by the handedness of the curvature of the chiral channels. This preference in twisting and folding in the direction of the winding of the helical channels is demonstrated by the average value of the writhe whose sign matches the writhe of the channel (Figure 3b). The channel-induced writhe was investigated in the previous work and we show the average writhe computed by evaluating Gauss integral [77] in Figure 3b. Twisting of the loops is also indispensable for creating knots by threading, as illustrated by the drawing in Figure 3c and demonstrated using a physical analog model (Appendix A) for which we used rubber tubes and 3D printed models. 

Additionally, we show the evolution of the writhe throughout the simulations in Appendix A. The graphs in Appendix A indicate that the writhe evolution begins with an abrupt change upon initiation of pushing, sometimes with a arbitrary direction of writhing as the pushing forces come into play. The average value and sign of the writhe further evolve, so that the sign aligns with the chirality of the channels. In the case of cylindrical channels, the sign tends to approach zero after a sufficiently long pushing inside the channels. These evolutions indicate that the handedness of knots is not inherited from a arbitrary initial structure after a long equilibration run, and that the process of controlling handedness benefits from a longer period of pushing inside the channels. Hence, the overall writhe of the chiral spaces plays a pivotal role in determining the chirality of the knots and influences their bias toward a particular chiral form. 

## 4. Conclusions

By means of coarse-grained molecular dynamics simulations, we studied the behavior of polymers in terms of polymer metrics, monomer distributions and topology of the polymer chain pushed inside infinite nanochannels with both cylindrical and helical geometries, using a DNA biopolymer model system. The simulations showed that the polymer undergoes a compaction upon increasing pushing velocity that could be used for controlling knottedness. 

When compared to simulations of polymers compressed in finite channels, distinct features emerge. Primarily, the geometry of the channels exerts varying effects on the extent of polymer chain compaction. Consequently, when the polymer is pushed inside open channels, it forms fewer knots compared to when it is compacted by compression within finite channels against an impenetrable wall. The confined environment of the open channels limits the polymer’s ability to explore helical loops of the chiral nanochannels, but it still generates equichiral knots and equichiral writhe. The lower degree of knotting observed in helical loops during the pushing inside helical channels may seem inconsistent with Ralf Metzler’s argument regarding enhanced knotting resulting from irregularities in nanochannels [78]. However, it is essential to note that the argument did not specify the spacing between these irregularities, whereas in our case, the irregularities are represented by the helical loops. Therefore, we posit that if the spacing between these irregularities is smaller than the DNA’s persistence length, their effects become translated into the increased confinement strength, which counteracts polymer folding.

Our findings also prompted additional simulations involving variations in the pitch of the channels and the radius of the helix. Some of these simulations were included in the discussion of the results and provided as Appendix A. They unveiled unprecedented backfolding and facilitated control over chirality, setting the stage for further investigation. The simulations demonstrate the feasibility of controlling polymer chirality by pushing them inside helical nanochannels, a feat achievable in experimental settings. This control over chirality can be fine-tuned by adjusting channel geometry, such as the pitch, the pushing regime (velocity, duration), channel length, and chain length. The last aspect, exploring chain lengths, remains a significant focus, especially in the context of wider channels. For a more comprehensive examination of knot spectra, simulation schemes employing Monte Carlo simulations, as developed in existing works, would be more suitable than molecular dynamics simulations. This approach allows for efficient conformational sampling at lower ligating distances, without generating correlated conformations, and minimizing bias induced by the closure method. Investigating knots at the ligating distance holds importance for potential applications such as chemical synthesis and embedding knots onto the polymer chain through methods like photoinitiated polymerization or click chemistry [79,80].

## Figures and Tables

**Figure 1 polymers-15-04185-f001:**
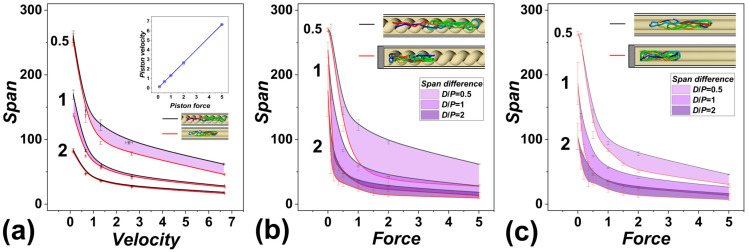
Polymer metrics in terms of the molecule’s span during pushing of DNA molecule inside channels as a function of pushing speed and confinement strength expressed as *D*/*P*. (**a**) The panel compares the polymer span in units of *σ*/ℓ pushed with different velocities of the piston *v* in units of 1000 *σ*/*τ*. The black lines indicate the span of a polymer pushed in helical channels, and red lines correspond to the simulations in a cylindrical channel. The areas in shades of purple show differences with the respective cases and a given confinement strength. The inset shows the velocities of the piston obtained for the external force employed on the piston. (**b**) The panel shows the span of polymer pushed in open infinite helical channels and compares the data with a previously investigated case of a polymer compressed in a blinded channel [36]. (**c**) The panel shows a comparison of the polymer’s span in open infinite channels versus blinded channels with cylindrical geometry and as a function of force in the units of *Fσ/ε*_0_ and confinement strength expressed in terms of *D*/*P*. The insets show rendered snapshots obtained for *D*/*P* = 0.5 and *F* = 5*ε*_0_/*σ*, where the polymer is shown in a rainbow color scheme, with one end of the polymer shown in blue and the other in red.

**Figure 2 polymers-15-04185-f002:**
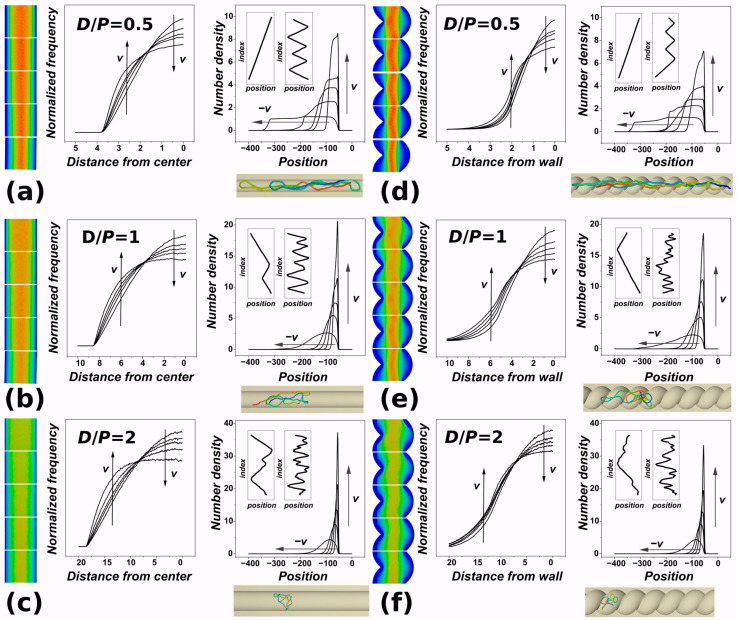
Monomer distributions of DNA polymer pushed inside channels with cylindrical and helical geometry. The panels show the distributions obtained in cylindrical (**a**–**c**) and helical channels (**d**–**f**). The distributions are also shown for different confinement strengths in terms of the *D*/*P* ratio, equal to 0.5 (**a**,**d**); *D*/*P* = 1 (**b**,**e**); and *D*/*P* = 2 in panels (**c**,**f**). The very left of each panel shows heatmaps of the monomer distributions with a color gradient ranging from blue to red, where the distribution’s peak is indicated by red and zero occurrences are denoted by blue. The left graph in every panel shows monomer distribution along the channel from the position (in units of *σ*/ℓ) of the piston as a function of piston velocity, taking into regard also direction of pushing in the simulations. The insets of this graph also show index-to-bead projections of axial monomer distributions obtained for *F* = 0.1*ε*_0_/*σ* and *F* = 5*ε*_0_/*σ*. The graphs to the right in the pair in every panel show radial distribution functions of monomers from the geometric center of the channel in units of *σ*/ℓ, in simulations represented by the *x* = 0 axis. The arrows indicate the direction of increasing or decreasing velocity. Each panel also shows a snapshot obtained for the particular geometry of the channel, confinement strength *D*/*P*, and force = 1 *ε*_0_/*σ*. The polymer is depicted in rainbow coloring, with the first bead in blue and the last bead in the chain in red.

**Figure 3 polymers-15-04185-f003:**
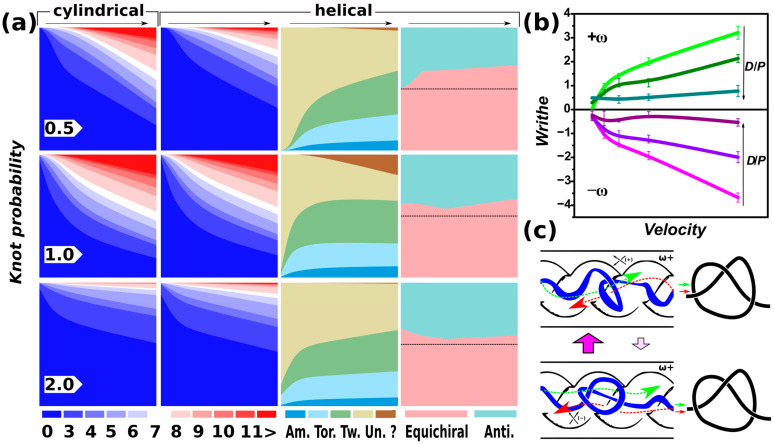
Knotting probabilities. (**a**) The first two columns compare knotting probabilities in terms of crossing numbers, distinguished by a thermometer scale, obtained for DNA polymer pushed inside infinite open channels with cylindrical and helical geometries. The rows correspond to different confinement strength in terms of *D*/*P*. In the case of helical channels, the knot types are evaluated in terms of the frequencies of amphichiral (Am.), torus (Tor.), twist knots (Tw.), unknots (Un.), and undefined knots (“?”). The last column shows a comparison of equichiral and antichiral knots, i.e., the knots with the same handedness as or the opposite handedness to that of the chiral helical channel. (**b**) Average writhe of the DNA chains in helical channels as a function of increasing pushing velocities and strength of confinement in terms of the *D*/*P* ratio. The *ω*+ and *ω*− indicate handedness of the channels (see Section 2.2). The computed pushing velocities are shown in the inset of Figure 1a. (**c**) A proposed mechanism for how the helical channels control handedness of compression–confinement-induced knotting.

## Data Availability

All data are available on request.

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
