# Peer review of "Knot Formation on DNA Pushed Inside Chiral Nanochannels"

_polymers, 2023, doi:10.3390/polym15204185_

Round 1
Reviewer 1 Report
1) Authors should expand their introduction section to explain what has been done (and found) in the literature and then carefully place their research motivation/results to allow readers to grasp the findings more easily. The introduction sections contain repetitive and vague sentences referencing the existing literature. Also, I am sure that knots (particularly DNA knots) are well-studied, requiring a more extensive literature review.
2) Besides "polymer span", the radius of gyration or the end-to-end distance can allow comparison with theoretical scaling models.
3) The chosen Kb bending strength does necessarily provide a persistence length of 50 nm (see Moreira and Kremer, 2015).
4) In all figures, axes labels are missing their units.
5) Were simulations performed in implicit solvent? If so, how can authors model/discuss the hydrodynamics? The Langevin dynamics may not provide a proper description for such systems, particularly in non-equilibrium conditions (Pastorino, PR E 2007).
6) The polymers are more shrunken as the velocity is increased. However, this is mainly because the time scale related to the pushing is shorter than the relaxation time of the polymer. It is also likely that the Langevin thermostat, which re-scales the velocities of monomers individually, may cause this effect. A DPD scheme can be tested to confirm these results.
7) Do the results have any N dependence? (see Virgiliis et al, JCP 2012).
Finally, some paragraphs may be divided into shorter sections to allow an easier reading experience.
n/a
Reviewer 2 Report

Inside "Comments and Suggestions for Authors
"
Round 2
Reviewer 1 Report
As a final check, the integration step tau= 0.01 is larger than the accepted standard. Even though the authors cite several previous works, they should provide evidence (in the SI text) that this timestep preserves the total energy (i.e., the system's energy does not drift away over time).
N/A
Author Response
Please, see attached.

Reviewer 2 Report
The authors have successfully addressed my previous issues/comments and I now fully support publication in Polymers
Author Response
We thank the Reviewer for his/her time to help us improve the quality of the manuscript.